# Low-Energy Electron Induced Reactions in Metronidazole at Different Solvation Conditions

**DOI:** 10.3390/ph15060701

**Published:** 2022-06-02

**Authors:** Christine Lochmann, Thomas F. M. Luxford, Samanta Makurat, Andriy Pysanenko, Jaroslav Kočišek, Janusz Rak, Stephan Denifl

**Affiliations:** 1Institut für Ionenphysik und Angewandte Physik and Center for Biomolecular Sciences Innsbruck, Leopold-Franzens Universität Innsbruck, Technikerstrasse 25, A-6020 Innsbruck, Austria; christine.lochmann@uibk.ac.at; 2J. Heyrovský Institute of Physical Chemistry of the Czech Academy of Sciences, v.v.i., Dolejškova 3, 18223 Prague, Czech Republic; thomas.luxford@jh-inst.cas.cz (T.F.M.L.); andriy.pysanenko@jh-inst.cas.cz (A.P.); jaroslav.kocisek@jh-inst.cas.cz (J.K.); 3Laboratory of Biological Sensitizers, Physical Chemistry Department, Faculty of Chemistry, University of Gdańsk, 80-308 Gdańsk, Poland; samanta.makurat@ug.edu.pl (S.M.); janusz.rak@ug.edu.pl (J.R.)

**Keywords:** metronidazole, radiosensitizer, low-energy electron, electron attachment, reduction, hydration, clusters

## Abstract

Metronidazole belongs to the class of nitroimidazole molecules and has been considered as a potential radiosensitizer for radiation therapy. During the irradiation of biological tissue, secondary electrons are released that may interact with molecules of the surrounding environment. Here, we present a study of electron attachment to metronidazole that aims to investigate possible reactions in the molecule upon anion formation. Another purpose is to elucidate the effect of microhydration on electron-induced reactions in metronidazole. We use two crossed electron/molecular beam devices with the mass-spectrometric analysis of formed anions. The experiments are supported by quantum chemical calculations on thermodynamic properties such as electron affinities and thresholds of anion formation. For the single molecule, as well as the microhydrated condition, we observe the parent radical anion as the most abundant product anion upon electron attachment. A variety of fragment anions are observed for the isolated molecule, with NO_2_^−^ as the most abundant fragment species. NO_2_^−^ and all other fragment anions except weakly abundant OH^−^ are quenched upon microhydration. The relative abundances suggest the parent radical anion of metronidazole as a biologically relevant species after the physicochemical stage of radiation damage. We also conclude from the present results that metronidazole is highly susceptible to low-energy electrons.

## 1. Introduction

Nitroimidazoles (NIs) have developed a broad spectrum of applications in human and veterinary medicine [1,2]. Since their development in the middle of the last century, they have been used mainly as antimicrobials. NIs are so-called pro-drugs, i.e., they develop their toxic action only after a specific activation process [3]. In the case of NIs, it was suggested that the enzymatic reduction of the nitro group leads to their activation [4]. However, this activation process turned out to be relevant only under anaerobic conditions since in aerobic environments the excess charge received is transferred further to molecular oxygen and the drug becomes deactivated [5].

The high selectivity in drug activation depending on the oxygen concentration also provided promising oncologic applications such as tumor imaging and radiotherapeutic treatment [6]. Due to high metabolism, solid tumors develop regions of low oxygen concentration. Such regions containing hypoxic tumor cells are characterized by reduced sensitivity towards ionizing radiation since molecular oxygen is considered to be an intrinsic radiosensitizer [7]. In radiotherapy, NIs are considered to mimic the effect of oxygen, though their cellular action is not well understood yet. However, like in the treatment against bacterial infections, reduction processes were considered to be highly relevant. Nevertheless, it has not been established yet which is the cytotoxic agent—the parent radical anion or its molecular fragments [2]. Several radio-oncologic trials with the three most important NIs, nimorazole, metronidazole, and misonidazole, turned out to be successful so far for nimorazole only, while the other two molecules showed limiting side effects at the required drug dose for radiotherapy [8,9]. Thus, new methods for the efficient transport of the radiosensitizers to the hypoxic tumor cells are investigated [10], in order to reduce the cytotoxic effects for healthy tissue.

Due to the activation by reduction, it may be concluded that the observed efficiency in medical treatment is based on a certain chemical property—the electron-affine nature of NIs. The basic imidazole ring has a negative electron affinity [11], i.e., the transient negative ion (TNI) formed by the attachment of an electron is energetically above the ground state of the neutral molecule. Substitution with the NO_2_ group at either the C2 or C5 position of the imidazole ring leads to a positive electron affinity, i.e., a bound state of the TNI then exists [12]. The electron affinity of nitroimidazolic compounds such as nimorazole and misonidazole is about 1.3 eV [13]. Hence, the capture of an electron with kinetic energies of almost zero eV leads to the vibrational excitation of their TNIs by the amount equal to the electron affinity. For the isolated molecule, only the sufficient intramolecular vibrational redistribution of this energy may lead to the formation of a long-lived parent anion with lifetimes > µs, as observed for nimorazole [14] and misonidazole [15,16]. In the case of electron attachment at kinetic energies > 0 eV, it becomes much more likely that the redistribution of energy is too inefficient to compete with the spontaneous emission of the excess electron or the dissociation of the TNI. The latter may occur if dissociation channels are energetically opened [17]. For example, for nimorazole, 15 different dissociative electron attachment (DEA) channels were found within the detection limit of crossed electron/molecular beam experiments [14,18].

The electron-induced chemistry in a molecule is commonly expected to be modified upon solvation [19,20]. This effect can be mainly ascribed to the possibility of the intermolecular redistribution of excess energy, which leads to the stabilization of TNI. Additionally, the caging effects by surrounding water molecules may occur [14,21]. Then, a question occurs if particular dissociation channels are still present for the solvated molecule or new dissociation paths open up due to the presence of the solvent. The nimorazole and misonidazole molecules showed opposite tendencies upon microhydration. Namely, the quenching of fragmentation channels upon microhydration is observed for nimorazole [14]. In contrast, for misonidazole the OH^−^ channel remained open under microhydration [15], and this channel was found to be independent of the hydration stage within the statistical uncertainties. Quantum-chemical studies proposed ring formation in the neutral radical fragment along with the release of OH^−^ [15].

In the present work, we investigated electron attachment to the NI molecule, metronidazole (Metro), which has not been studied in the context of its electron attachment properties so far. The chemical structure of Metro is shown in Figure 1. We studied the electron induced chemistry in two different settings: the isolated molecule and the microhydrated molecule in small clusters. The former condition provides information on the basic properties of Metro regarding the attachment of a low-energy electron. The latter setting may be considered as a first approach to study the solvation effects in the TNI formed by attachment of a free electron e^−^_f_. In an aqueous solution, such secondary electrons formed by the interaction of high-energy radiation with biological matter have a most probable energy of about 9–10 eV in a rather broad distribution from 0 eV up to more than 100 eV [22]. These low-energy electrons are quickly thermalized by elastic and inelastic collisions and enter the so called pre-hydrated stage e^−^_pr_, where they exist in shallow potential traps in the range of zero to −1.6 eV [23]. Bound in the water matrix by at least 1.6 eV, electrons enter the stage of a fully solvated electron e^−^_aq_, which is also chemically reactive [23,24].

## 2. Results and Discussion

The attachment of an electron e^−^ to a molecule M, irrespective of its solvation status, leads in the first step to the formation of the TNI (M_TNI_)^−^ *,
M + e^−^ → (M_TNI_)^−^ * → A^−^ + B(1)

The formation of the TNI is an ultrafast reaction, occurring on a femtosecond timescale, and a resonance process, i.e., its efficiency, which in scattering physics is also expressed by a cross section, strongly depends on the initial electron energy [25,26,27]. As pointed out in the Introduction, three relevant pathways exist for the decay of the TNI: (i) the stabilization of the TNI by sufficiently fast energy dissipation, (ii) spontaneous electron emission, and (iii) the DEA process indicated in the second step of Equation (1) leading to a charged fragment anion A^−^ and neutral fragment(s) B [26]. The mass spectrometry applied here provides direct information about the charged reaction products, i.e., the result of (i) and/or (iii).

Due to the resonance nature of Reaction (1), mass spectra at different kinetic energies of the electron were recorded for the isolated molecule, in order to obtain a first overview about the variety of anions formed upon electron attachment. Figure 2 presents the overall negative ion mass spectrum of Metro. This spectrum was obtained by summing 15 individual mass scans recorded in 0.25 eV steps for the electron energy range between ~0 and 3.75 eV. In this energy range, abundant signals are found. The spectrum clearly indicates that the associative electron attachment process leading to the formation of the intact parent radical anion (mass 171 u) is by far the most abundant reaction upon the attachment of an electron to the isolated molecule. The parent anion yield is roughly one order of magnitude higher than that of the most abundant fragment anion. This indicates the exceptionally high stability of the parent radical anion against dissociation, though, as will be discussed below, the dissociation channels are energetically open at the electron energy of already zero eV. Figure 3 shows the energy-resolved anion efficiency curve for the undissociated parent anion. As expected, the formation of the parent anion is restricted to electron energies close to about zero eV.

We also estimated the electron attachment cross section for formation of the Metro parent anion, using the procedure mentioned in Section 3.1. We obtained a cross section of σ_Metro_ = 3.6 × 10^−18^ m^2^. We can directly compare this value with the previous determination of the associative electron attachment cross section of the nimorazole radiosensitizer using the same apparatus, σ_Nimorazole_ = 2.8 × 10^−18^ m^2^ [14]. Though the systematic uncertainty of these values is estimated to be one order of magnitude, both compounds have evidently a similar cross section for electron attachment. For comparison, we also recently determined the DEA cross section for the proposed radiosensitizer benzaldehyde and obtained maximal values on the order of only 10^−22^ m^2^ [28]. Hence, this extraordinary high cross section of Metro provides further insight into the action of NIs based radiosensitizers. As pointed out in [3], since the yield of secondary electrons is limited, when high-energy radiation interacts with water, only high electron affinity species may act as radiosensitizers since other endogenous electron acceptors are present in the cells. For both Metro and nimorazole, associative attachment is the main reaction channel leading to the parent radical anion. We calculated the adiabatic electron affinity (AEA) of Metro at the M06-2X/aug-cc-pVTZ level (Section 3.3) and obtained 1.24 eV. This value of the AEA is lower than the one previously reported (1.54 eV) [29].

A more detailed view at the ion signal in the mass region below the parent anion indicates that several dissociation channels are present, see Figure 2. Here, we will focus mainly on the most relevant species, which are also observable under microhydration. All other fragment anions, formed upon more complex reactions that also require rearrangement of bonds in the TNI, are presented in details in the Appendix A, including a table with a summary of experimental and calculated thresholds (Appendix A), suggested reaction schemes (Appendix A and the respective xyz coordinates), and anion efficiency curves (Appendix A). The most abundant fragment anion appearing in the mass spectrum is found at *m*/*z* 46, which can be assigned to the formation of the NO_2_^−^ anion. The quantum chemical calculations show that the formation of this anion is driven by the appreciable electron affinity of the NO_2_ fragment (2.35 eV), which leads to an exothermic reaction by −0.35 eV at 360 K. The anion efficiency curve of NO_2_^−^ is shown in Figure 3, indicating a peak near zero eV (~0.05 eV) that agrees with the proposed exothermicity of the DEA reaction. Another, less intense, peak is found at about 0.3 eV; it may be assigned to the activation of the symmetric stretching vibration mode of NO_2_ with an energy of 0.163 eV [30]. Besides a weak feature near 1.5 eV, the main resonance associated with the NO_2_^−^ ion can be found at about 3 eV. We note that the principal shape of the anion yield with its vibrational structure near zero eV, a feature at about 1.5 eV, and a broad peak ~3 eV is very similar to that found in DEA to nimorazole [14] and misonidazole [16]; however, the ratio of yields at ~0 eV and ~3 eV varies strongly between the compounds. The largest ratio is found in misonidazole (~4), followed by Metro (~0.4) and nimorazole (~0.03). This variation of the ratio may arise from subtle differences in the decay process of the TNI. The dissociation process leading to NO_2_^−^ is due to the simple cleavage of the C-NO_2_ bond. For the nitroimidazole molecules, Kossoski and Varella suggested an indirect mechanism for the dissociation process, in which the coupling of initial π* resonances of the nitroimidazole molecule and the repulsive σ* CN states occurs [31]. Slight differences in the lifetimes of the intermediates and the coupling rates may easily influence the observed anion yields for the different compounds. It is worth mentioning that a very similar peak structure at ~1.5 and ~3 eV was also observed in NO_2_^−^ ion yield upon DEA to 2-nitrofuran [32]. This indicates that the suggested coupling mechanism may be active irrespectively of the composition of the ring or the side group in the nitro compound.

As pointed out in the Introduction, since there is a notable difference between nimorazole and misonidazole concerning OH^−^ formation, we also pay attention to this reaction channel in the case of Metro. The corresponding anion efficiency curve for OH^−^ is shown in Figure 3. The anion is formed with minor abundance and the overall shape of the ion yield is similar to that of NO_2_^−^, though the features near zero are less resolved and the maximum of the main peak near 3 eV is slightly shifted to higher electron energies (~3.2 eV). To form this fragment anion, one may assume a simple C-O bond cleavage within the side chain as the most obvious dissociation process. However, the calculated threshold for such a simple bond cleavage is endothermic by +1.16 eV, which is not in agreement with the features near zero eV and, thus, suggests a complex dissociation process. For instance, a rearrangement reaction (see Appendix A), where additionally the bond between the ring and the side chain is broken and the NO_2_ group has exchanged its position with a hydrogen atom of the side chain, would lead to an exothermic reaction (−0.74 eV). For the more intense ion yield peaking at about 3.2 eV, all of the considered reaction channels are open. We just note that close to 3–4 eV, most fragment anions show an intensity maximum, see SI. Thus, the corresponding TNI state, suggested as π*_3_ resonance of the imidazole moiety [31], is an essential precursor state in the formation of fragment anions from Metro.

Moving now to the micro-hydrated conditions, the corresponding negative ion mass spectrum derived as a sum of 20 individual mass spectra in the electron energy range between about 1 and 6 eV (recorded in steps of 0.25 eV) is shown in Figure 4. The region below 1 eV is not included due to the instabilities of the electron source at lower energies. The average number <*n*> of water molecules attached to the Metro anion is about 3 per metronidazole molecule for this measurement, which was calculated from the intensities of the Metro(H_2_O)*_n_*^−^ mass peaks in the spectrum (for details see Ref. [33]). Since water molecules may evaporate from the cluster upon electron attachment, <*n*> is lower but directly proportional to the mean number of water molecules in the neutral precursor cluster. The abundance of the peaks above the mass of the Metro parent anion indicates the successful solvation of the parent anion by water molecules. In this case, an excess electron is attached to hydrated Metro and the excess energy (comprised of the electron affinity of Metro and the kinetic energy of the electron) will first be distributed among the intra- and inter-molecular degrees of freedom within the cluster. This process may further lead to the evaporation of a number of water molecules from the cluster, which further stabilizes the cluster and reduces the cleavage of intramolecular bonds. An inspection of the mass region below the parent anion indicates that the intensity of mass peaks is rather low compared to the higher mass region. Few peaks can be also considered as background, see Figure 4. For example, the mass peaks at *m*/*z* 79 and 81 result from electron attachment to a residual background containing bromine. We also assign the mass peaks in the mass range between *m*/*z* 127 and *m*/*z* 168 to background signals. At the solvation conditions chosen, the ion signal of NO_2_^−^ (*m*/*z* 46) is observable. To evaluate if a quenching effect on the dissociation process occurs, we determined the ratio of NO_2_^−^ ion yield and ion yield associated with the intact Metro anion (bare and hydrated) for different hydration conditions, see Figure 5. This figure indicates a clear solvation effect, i.e., the abundance of the dissociation channel becomes reduced with increasing microhydration. The determined ratios in Figure 5 can be compared with those obtained for nimorazole and misonidazole, determined with the same experimental setup. Comparing the ratio at approximately the same hydration condition with a mean number of about three water molecules surrounding the radiosensitizer, the reduction of the NO_2_^−^ dissociation is about a factor of ~70 for Metro and thus closer to nimorazole (~100 [14]) than to misonidazole (~10 [15]).

In the cluster mass spectrum shown in Figure 4, weakly abundant anion yield is also present at *m*/*z* 123. For the isolated molecule, we also observe ion yield at this *m*/*z* ratio, see Figure 2. We assign this ion yield to (Metro-NO_2_-2H)^−^, i.e., the NO_2_ group and two hydrogen atoms (or alternatively a H_2_ molecule) were dissociated from the TNI. The ratio to the NO_2_^−^ yield is similar to that for the isolated molecule, i.e., this channel also becomes strongly quenched in the cluster. This effect becomes even more evident for the other heavier fragment anions in the spectrum in Figure 2. Except (Metro-NO_2_-2H)^−^, those fragment anions are below the detection limit in the cluster environment. The complexity of the dissociation process (multiple bond cleavage including rearrangement) may serve as an explanation for this effect since the energy-dissipating environment can act as an effective energy sink preventing complex dissociation processes [34,35], and, in addition, caging by the surrounding water molecules may sterically block rearrangement processes [34,36]. A different tendency is observed for the OH^−^ fragment anion. While this fragment anion has very low abundance in the case of the isolated molecule (~1/10 of the NO_2_^−^ ion yield), the intensities of OH^−^ and NO_2_^−^ are very similar for clusters with a ratio of ~4/5. Moreover, the mass spectrum shown in Figure 4 also shows the OH^−^ anion clustered with one water molecule (indicated by a peak at *m*/*z* 35), while the NO_2_^−^ fragment anion is only observed in its bare form. Thus, the tendency for OH^−^ formation upon microhydration is similar to that one observed for misonidazole [15]. The linear side chain (allowing facile rearrangement) is crucial for this effect, as well as the presence of an OH group in this chain, making single-bond cleavage sufficient for the abstraction of the anionic species in the water environment, in contrast to isolated molecule conditions where a complex rearrangement is needed (see Appendix A).

## 3. Methods

### 3.1. Experiments with Isolated Metro

The electron attachment experiment with isolated Metro was performed with a crossed electron-molecule beams apparatus previously described in more detail [14]. The device, located at the University of Innsbruck, consists of a resistively heated oven to generate a molecular beam of Metro, a high-resolution hemispherical electron monochromator (HEM), a quadrupole mass filter, and a detection system for the anions. Briefly, the solid Metro sample (purchased from Sigma-Aldrich, purity 99.99%) was used as delivered and evaporated in the oven at about 360 K inside the vacuum chamber. The molecules in the gas phase were introduced via a capillary of 1mm diameter into the interaction region with the electron beam. The anions formed by electron attachment were extracted by a weak electrostatic field into the quadrupole mass filter, and, after mass analysis, transmitted anions were detected by a channeltron type secondary electron multiplier. The well-defined electron beam was generated by the HEM. The measurements were carried out at the energy beam resolution of about 120 meV full-width-half-maximum (FWHM) at an electron current of about 25 nA, which represents a reasonable compromise between energy resolution and intensity. The determination of the FWHM and the calibration of the electron energy scale were performed with the well known SF_6_^−^ resonance formed closely at the electron energy of zero eV upon electron attachment to SF_6_.

In this study, we recorded mass scans at fixed electron energies as well as anion efficiency curves at variable electron energies for a selected mass. From the latter data, the energy onsets as well as the peak maxima for the corresponding detected resonances were determined by Gaussian fits. For illustration purposes only, the cumulative Gaussian curve is provided in the anion efficiency curves shown. The peak thresholds x_thres_ were determined by subtracting the 2σ range from the center x_c_ of the Gaussian fit [37] (2):x_thres_ = x_c_ − 2σ(2)

We also provide an estimation of the electron attachment cross section of Metro, which was determined by comparing the Metro ion yield with the well-known cross section of Cl^−^/CCl_4_ at 0.8 eV (5.0 × 10^−20^ m^2^ [38]). In this procedure, the experimental conditions were considered by normalizing the anion signal intensity with respect to the partial pressures.

### 3.2. Experiments with Microhydrated Metro

Metro was clustered with water molecules in the CLUster Beam (CLUB) apparatus at the Czech Academy of Sciences in Prague. More detailed information about this setup can be found in Refs. [34,39]. Coexpansion with a humid gas was used for cluster production [40]. In the first step, helium or neon gas was humidified by a Pergo gas humidifier. The latter is based on a Nafion tubing gas line that passes through a water bath, and its membrane selectively permeates water. The humidified gas was introduced into a heated oven filled with Metro. The oven temperature used was about 400 K. After the oven, the mixture of the humidified buffer gas and the vapor of Metro was co-expanded through a conical nozzle (opening diameter 90 μm). The seeding gas pressures varied between 1.5 and 3 bar, which led to Metro(H_2_O)*_n_* clusters with different average sizes. The cluster beam was skimmed after a distance of ~2.5 cm and crossed by an electron beam in the reflectron time-of-flight mass analyzer with a mass resolution of ~5∙10^3^. For each extraction pulse, all anions were mass analyzed and finally detected by a multichannel plate.

### 3.3. Quantum Chemical Calculations

The thresholds of DEA reactions in Metro were calculated for the most stable conformer of metronidazole (shown in Figure 1) as the difference between the Gibbs free energies, ΔG of reactants in their ground state, see Equation (3) below. In this equation, G_p,T_ is the free enthalpy at the pressure p and temperature T. For all the calculations, we used the M06-2X [41]/aug-cc-pvtz [42,43] method/basis set combination and the Gaussian09 [44] suite of programs. The thermochemical characteristics for the standard state (298.15 K, 1 atm), as well as the experimental conditions (360.15 K, 2.92 × 10^−11^ atm), were computed. The pressure correction to the G value for the experimental pressure was obtained using Equation (4) [45], in which S_trans;p,T_ denotes translational entropy at the pressure p and temperature T. This approach was previously shown to be successful [46,47,48,49].
(3)$${\Delta G}_{p,T}=G_{p,T}\left(\mathrm{products}\right)-G_{p,T}\left(\mathrm{Metro}\right)\;$$
(4)$$G_{3x10^{-11}\mathrm{atm},T}=G_{1\mathrm{atm},T}+{\;\mathrm{TS}}_{\mathrm{trans};1\mathrm{atm},T}-{\;\mathrm{TS}}_{\mathrm{trans};\;3x10^{-11}\mathrm{atm},T}\;$$

Additionally, the adiabatic electron affinity (AEA) was calculated for all anionic species, based on Equation (5):(5)$$\mathrm{AEA}\;=G_{\mathrm{neut},\;\mathrm{geom}:\mathrm{neut}}-G_{\mathrm{anion},\;\mathrm{geom}:\mathrm{anion}\;}$$

## 4. Conclusions

In the present work, we investigated low-energy electron attachment to the potential radiosensitizer, metronidazole, at isolated condition and upon microhydration. The latter acts as a first approach to mimic a more complex solvated stage. The present results show that the metronidazole parent radical anion is formed with high cross section, while its dissociation becomes quenched upon microhydration. The only exception is found for OH^−^ formation, which is enhanced in the water clusters. However, its abundance compared to the (hydrated) parent radical anion is minor, and thus the parent radical anion of metronidazole seems to be the only relevant species formed by the single electron reduction in the metronidazole molecule. Such a conclusion would be in line with previous pulsed radiolysis studies of metronidazole in the solution phase and its suggested radiobiological action [50]. This aspect may also point out the relevance of molecular studies on small-scale models for the future development of potential radiosensitizers, as suggested in [51].

## Figures and Tables

**Figure 1 pharmaceuticals-15-00701-f001:**
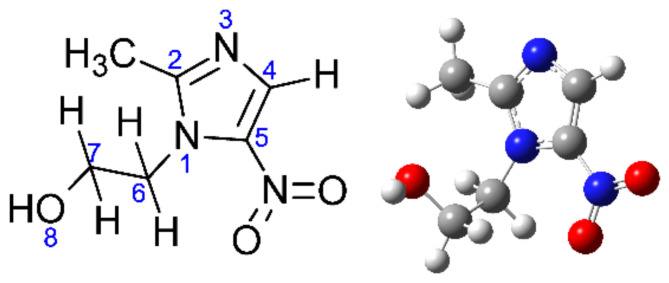
The structure of metronidazole with atom numbering (left) and a 3D structure (right).

**Figure 2 pharmaceuticals-15-00701-f002:**
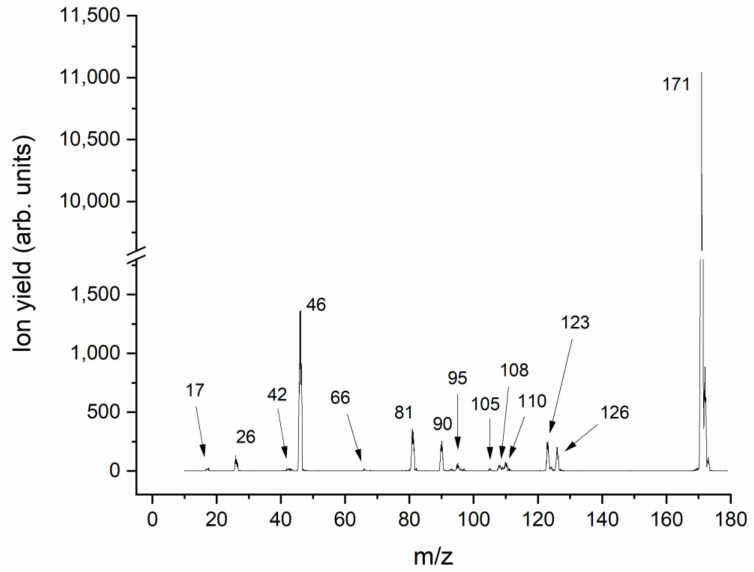
The negative ion mass spectrum of isolated Metro obtained by the sum of single mass spectra in the electron energy range ~0–3.75 eV in 0.25 eV steps. The mass-to-charge ratio of anions further studied by the measurement of the corresponding anion efficiency curves and quantum chemical calculations are mentioned.

**Figure 3 pharmaceuticals-15-00701-f003:**
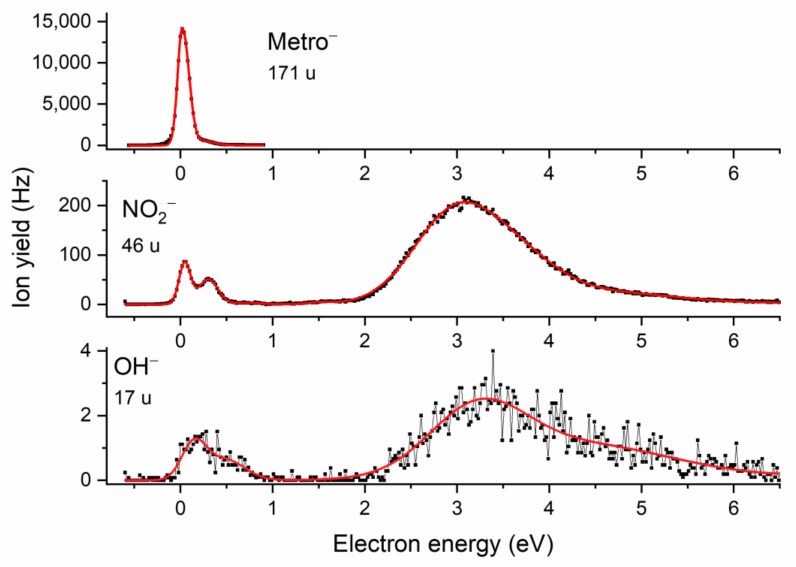
The anion efficiency curves for the Metro parent radical anion Metro^−^ (top), NO_2_^−^ (middle), and OH^−^ (bottom) upon electron attachment to isolated Metro.

**Figure 4 pharmaceuticals-15-00701-f004:**
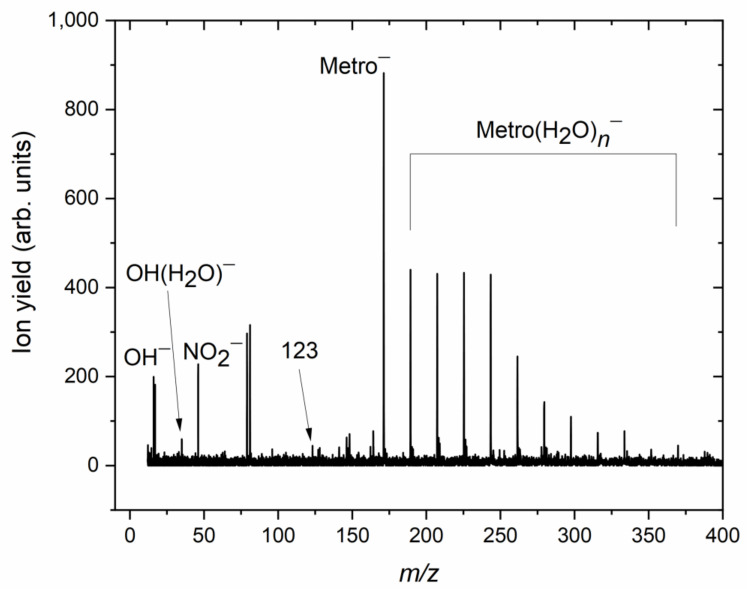
The negative ion mass spectrum for Metro clustered with water. The spectrum was obtained by the sum of individual mass spectra recorded in the electron energy range between about 1 and 6 eV in steps of 0.25 eV. The spectrum indicates the parent anion of Metro clustered with *n* = 1–11 water molecules. The peak at mass-to-charge 123 is assigned to (Metro-NO_2_-2H)^−^. The other mass peaks not labeled represent background signals.

**Figure 5 pharmaceuticals-15-00701-f005:**
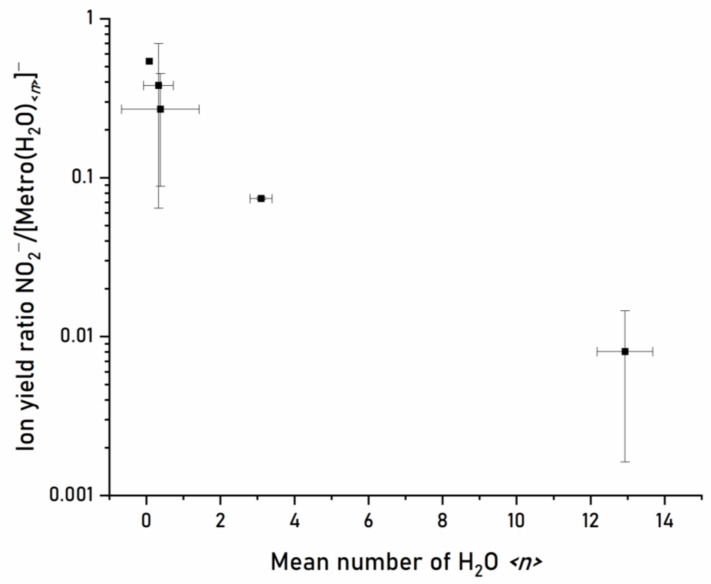
The ratio of the total yields of NO_2_^−^ and Metro(H_2_O)*_n_*^−^ parent anion at different hydration conditions, determined by the mean number <*n*> of water molecules attached to the parent anion in Metro(H_2_O)*_n_*^−^ clusters. The statistical uncertainty described by the standard deviation is indicated by the error bars.

## Data Availability

The raw data are included in the Appendix A of this article.

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
