# Peer review of "Low-Energy Electron Induced Reactions in Metronidazole at Different Solvation Conditions"

_pharmaceuticals, 2022, doi:10.3390/ph15060701_

Round 1

Reviewer 1 Report

The Authors have studied of electron attachment to metronidazole using two crossed electron/molecular beam devices with mass-spectrometric analysis of the formed product anions. The manuscript deserves to publish in Pharmaceuticals after a minor correction. I would like to suggest introducing changes before publishing in Pharmaceuticals.

The authors should revise in the manuscript as the following points:

  1. Abstract: The abstract should state briefly the purpose of the research, the principle results and major conclusions. The abstract should be corrected.
  2. Please correct equation 3.
  3. Please introduce structures in the supplement so that they can be recreated.
  4. Double-check the correctness of the text with the template.

Author Response

We thank Reviewer 1 for his/her favorable report. The comments helped us to further improve the quality of the manuscript. Our response addressing the comments are presented below. Please note that the detailed changes can be followed in the docx-file of the manuscript due to the used “Track-changes” function.

The Authors have studied of electron attachment to metronidazole using two crossed electron/molecular beam devices with mass-spectrometric analysis of the formed product anions. The manuscript deserves to publish in Pharmaceuticals after a minor correction. I would like to suggest introducing changes before publishing in Pharmaceuticals.

The authors should revise in the manuscript as the following points:

  1. Abstract: The abstract should state briefly the purpose of the research, the principle results and major conclusions. The abstract should be corrected.

Answer: We substantially revised the abstract in order to match it with the required structure.

  1. Please correct equation 3.

Answer: We removed the track-mode comment for this equation which was a leftover of the manuscript preparation.

  1. Please introduce structures in the supplement so that they can be recreated.

Answer: We added the xyz coordinates for Metro and all the products shown in Figures S1-S4 to Supplementary Material. A wording “and the respective xyz coordinates” has been added to the text of the manuscript (see p. 5, line 25).

  1. Double-check the correctness of the text with the template.

Answer: We checked the manuscript and placed few corrections followable in track-mode.

Reviewer 2 Report

The authors present a combined experimental and theoretical study of electron attachment to metronidazole (Metro) and the subsequent dissociation reactions.

The study of interactions of low energy electrons (LEE) with Metro is important since along with other nitroimidazoles it is a potential radiosensitizer for radiation therapy. Here LEEs are the main actors in inducing lesions in biological tissue. The authors use molecular gas targets crossed by monochromatized electron beams and the negatively charged molecular reaction products are mass analyzed. They measure ion production curves as function of the electron energy and they calculate the energetics of various fragmentation channels like the formation thresholds. As consequence they can conclude on the details of the fragmentation reactions. Furthermore, the present results for Metro are compared to published data from other nitroimidazoles. The particular strength of the present experimental work is the comparison of data for isolated and for hydrated molecules such that the influence of solvation can be directly identified. The authors conclude that hydration is quenching the dissociation with the only exception for the OH- formation.

The study is correctly designed and technically sound. The data shown are of high quality such that robust conclusions can be drawn. The work is within the scope of the journal and of strong interest to the readers. The article is well written and all relevant information and references are given. From these reasons I clearly support publication with some minor corrections.

Suggestions:

As far as I can see the calculations refer to the isolated molecule only. It would be helpful to have calculations of the cluster geometry for the neutral and the negative ion. It is interesting to see where the water binds to Metro. Is there rearrangement after electron attachment?

Some corrections:

  • The paragraph numbering is incorrect. All numbers are set to 1.
  • Page 3, second paragraph: “ … most probable energies of about 9-10 eV.” This is misleading since it is implied that the energy range of secondary electrons is narrow while in reality it is broad and ranging from 0 to beyond 100 eV.
  • Page 3, last paragraph: “The parent ion yield is about two orders of magnitude higher ….” According to the vertical scale in Fig. 2 it looks like one order of magnitude.
  • Page 6 last paragraph. “<n> is lower, but directly proportional …” I do not understand this sentence. Does it refer to <n> of the negative ion clusters? Please write more clearly.

Author Response

We thank Reviewer 2 for his/her favorable report. The comments helped us to further improve the quality of the manuscript. Our response addressing the comments are presented below. Please note that the detailed changes can be followed in the docx-file of the manuscript due to the used “Track-changes” function.

The authors present a combined experimental and theoretical study of electron attachment to metronidazole (Metro) and the subsequent dissociation reactions.

The study of interactions of low energy electrons (LEE) with Metro is important since along with other nitroimidazoles it is a potential radiosensitizer for radiation therapy. Here LEEs are the main actors in inducing lesions in biological tissue. The authors use molecular gas targets crossed by monochromatized electron beams and the negatively charged molecular reaction products are mass analyzed. They measure ion production curves as function of the electron energy and they calculate the energetics of various fragmentation channels like the formation thresholds. As consequence they can conclude on the details of the fragmentation reactions. Furthermore, the present results for Metro are compared to published data from other nitroimidazoles. The particular strength of the present experimental work is the comparison of data for isolated and for hydrated molecules such that the influence of solvation can be directly identified. The authors conclude that hydration is quenching the dissociation with the only exception for the OH- formation.

The study is correctly designed and technically sound. The data shown are of high quality such that robust conclusions can be drawn. The work is within the scope of the journal and of strong interest to the readers. The article is well written and all relevant information and references are given. From these reasons I clearly support publication with some minor corrections.

Suggestions:

As far as I can see the calculations refer to the isolated molecule only. It would be helpful to have calculations of the cluster geometry for the neutral and the negative ion. It is interesting to see where the water binds to Metro. Is there rearrangement after electron attachment?

Answer: Yes, indeed, our calculations address the isolated molecule. Although at the first glance the Referee’s remark seems to pose a relatively simple query, to give a solid answer one would have to conduct a whole new project. In order to properly tackle the microsolvation problem, quantum chemical characteristics for the Metro…(H2O)n clusters are needed. As indicated by Figure 5 one would have to calculate clusters comprising up to 13 molecules of water in order to discuss water binding and the observed quenching of NO2release.

Unfortunately, a microsolvation problem is not a trivial task. It is important to note that microsolvation is so called an NP-hard problem, meaning that time needed to solve it scales exponentially with its size. Thus, computer time necessary to obtain a solution becomes very quickly (already for systems of relatively small size) larger than any acceptable value. This is why some of us proposed recently a general-purpose computational approach which is a combination of molecular dynamics simulations followed by clustering of similar configurations and their quantum chemical refinement (J. Phys. Chem. Lett. 2022, 13, 3230−3236). Although, the mentioned above approach makes the NP-hard problem solvable, it still requires a sizeable amount of time related mainly to QM calculations – the project described in J. Phys. Chem. Lett. costs ca. half a year despite the fact that the analyzed clusters comprised up to 4 water molecules only. In the present work, we employ quite a large aug-cc-pVTZ correlation consistent tiple-zeta basis set. Additionally, taking into account the scaling of DFT calculations with system size and time needed for geometry optimization and frequency run for the Metro anion (equal to 850 min on 24-core Intel Xeon E5-2670 v3 2.3 GHz processor), one can calculate that going from Metrothrough [Metro…(H2O)3] to [Metro…(H2O)13](see Figure 5) requires  850, 2140 and 16680 min, respectively. Please, note that the time for [Metro…(H2O)­13]  amounts to ca. 12 days of continuous computer work. Remembering that the queues in computer centers (the resources are shared among several hundreds of users) effectively slow down calculations such a project could easily last a month or so. The above times needed to complete the calculations were estimated assuming the analysis of just three structures. One should realize, however, that due to a large structural flexibility of the hydrogen bonded clusters, the number of geometries that should be calculated can easily approach dozens rather than just one conformer for a given number of waters. Hence, a statistically sound description of microsolvation effects in the [Metro…(H2O)n] anions requires the execution of several-month project. We, therefore, decided against such calculations for the current paper.

Nevertheless, in order to partially address the Referee‘s remark, we would like to stress that Metro undoubtedly forms clusters with water - see Figures 4 and 5 depicting the result of our microsolvation experiments. Metro contains several proton-acceptor sites: N3, N1, O8, and the NO2 group which may interact with waters in the respective Metro…(H2O)n clusters. As a radical anion Metro forms even stronger hydrogen bonds then its neutral form since the partial charges on particular proton-acceptor centers increase due to presence of the excess electron. This effect expresses itself in the increase of electron affinity for hydrogen bonded forms. Indeed, the nucleobase valence anions’ adiabatic electron affinities small or even negative in the gas phase increase by even 2 eV in an aqueous solution (see e.g. Chem. Rev. 2012, 112, 5603−5640). On the other hand, the quenching effect (see Figure 5) is probably related to the increase of activation barrier for the C5-N bond cleavage in the radical anion. While the C5-N bond brakes, the excess electron is transferred from the imidazole moiety to the σ* C5-N orbital. Hydrogen bonds make the binding of excess electron stronger (the formation of the hydrogen bonded Metro anion is facilitated in comparison to the isolated molecule) which hinders transfer of the excess electron from the imidazole moiety to the σ* C5-N orbital. Already such simplistic reasoning seems to qualitatively explain the quenching mechanism of water molecules (Figure 5). Unfortunately, a rigorous approach, based on cluster calculations is, as indicated above, beyond the reach of the current paper.          

Some corrections:

  • The paragraph numbering is incorrect. All numbers are set to 1.

Answer: This problem raised during the conversion into the journal´s template. We revised accordingly.

  • Page 3, second paragraph: “ … most probable energies of about 9-10 eV.” This is misleading since it is implied that the energy range of secondary electrons is narrow while in reality it is broad and ranging from 0 to beyond 100 eV.

Answer: We revised accordingly and mention now the broad range of the distribution as well.

  • Page 3, last paragraph: “The parent ion yield is about two orders of magnitude higher ….” According to the vertical scale in Fig. 2 it looks like one order of magnitude.

Answer: The reviewer is right and we corrected this mistake accordingly.

  • Page 6 last paragraph. “<n> is lower, but directly proportional …” I do not understand this sentence. Does it refer to <n> of the negative ion clusters? Please write more clearly.

Answer: Yes, <n> refers to the negative ion clusters. This mean size can be deduced from the measured spectra, while the neutral cluster size is not accessible due to possible evaporation of water molecules upon TNI formation. In the revised version, we point out this situation by writing “Since water molecules may evaporate from the cluster upon electron attachment, <n> is lower,…”. In addition, we revised incorrect statements about <n> in the sentence before this one mentioned by the reviewer as well as in Section “3.2. Experiments with microhydrated Metro”.